# Multi-Point Deformation Prediction Model for Concrete Dams Based on Spatial Feature Vector

**Zhuoxun Chen * and Xiaosheng Liu**

School of Architecture and Surveying Engineering, Jiangxi University of Science and Technology, Ganzhou 341000, China; lxs9103@163.com
* Correspondence: chenzx524@outlook.com

**Abstract:** Deformation can effectively reflect the structural state of concrete dams and, thus, establishing na accurate concrete dam deformation prediction model is important for dam health monitoring and early warning strategies. To address the problem that the spatial coordinates introduced in the traditional multi-point deformation prediction model of dams not being able to accurately and efficiently reflect the spatial correlation of multiple-measuring points, a 2D-1D-CNN model is proposed which expresses the spatial correlation between each measuring point through spatial feature vectors, replacing the spatial coordinates in the traditional multi-point model. First, the spatial feature vector is extracted from the historical spatio-temporal panel series of deformation values of measuring points via a Two-Dimensional Convolutional Neural Network (2D-CNN); second, the vector is combined with the environmental impact factor of dam deformation to form the final input factor of fused spatial features; and, thirdly, this vector is combined with the environmental impact factors of dam deformation to form the final input factor of fused spatial features, and the non-linear linkage between the factors and the measured displacement values is constructed by the efficient feature processing capability of a One-Dimensional Convolutional Neural Network (1D-CNN) to obtain the prediction results. Finally, the actual monitoring data of a concrete dam in China are used as an example to verify the validity of the model. The results show that the proposed model outperforms the other models in most cases, respectively, which verifies the effectiveness of the proposed model in this paper.

**Keywords:** concrete dam; deformation prediction; convolutional neural network; deep learning





## 1. Introduction

Following the development of engineering technology, the number of dams in the world has gradually increased; meanwhile, the risk of dam failure has also been increasing. Deformation is the effect quantity that best reflects the operational safety status of dams. By analyzing the historical monitoring data of dams and establishing an accurate dam deformation prediction model, we can intuitively grasp the safety status of dams, which is important to ensure the stable operation of dams and is one of the research hotspots in the field of dam safety monitoring [1–3].

Currently, the commonly used dam deformation prediction methods can be classified into deterministic models, statistical models, and machine learning models. Deterministic models combine the actual structure and physical parameters of dams and dam foundations through the finite element method to obtain the evolution of dam deformation, which has high prediction accuracy and interpretability, but has the problem that some parameters are difficult to determine. The traditional statistical regression model is highly interpretable and has high operational efficiency, but it has insufficient ability to fit nonlinear relationships and is susceptible to multi-collinearity among its factors, resulting in poor model fitting [4]. Therefore, machine learning models that are more suitable for nonlinear relations and have stronger generalization ability are gradually introduced into the field

of dam deformation prediction, such as Su et al. [5], who combined wavelet analysis, a particle swarm optimization algorithm and support vector machine (SVM) to construct a ReP-WSVM model for dam deformation prediction, which has a greater improvement in prediction accuracy compared with the traditional SVM model. Kang et al. [6] proposed a Hydrostatic-Air temperature-Time ($HT_AT$) concrete dam prediction model based on Kernel Extreme Learning Machines (KELM); Wei et al. [7] developed a dam deformation prediction model considering chaotic residuals using a BP neural network (BPNN) and frog jump algorithm. Compared with the statistical regression model, the machine learning model can better extract the nonlinear relationship between the causal factors and the deformation of the measuring point, and improve the prediction accuracy [8].

In addition, most of the traditional dam deformation prediction models only model the deformation of a single measuring point and do not consider the spatial correlation of the deformation of different measuring points, which makes it difficult to reflect the overall deformation state of the dam [9]. As the study progressed, the traditional single-point model is gradually developed into a multi-point model that can take into account the spatial and temporal features. For example, Li et al. [10] proposed a deterministic multi-point model for concrete arch dams using the finite element method; Song et al. [11] proposed a wavelet-based SSA-ELM dam deformation spatio-temporal prediction model to address the problem that multi-point statistical models are susceptible to multiple covariance and noise; Wei et al. [9] combined the Finite Element Method(FEM) and SVM of particle swarm optimization to construct a hybrid multi-point model for concrete dams.

Most of the existing multi-point models express the spatial characteristics between measuring points by introducing the spatial coordinates of measuring points, which has the following main problems: (1) Due to the complex causes of dam deformation, the spatial distribution pattern of the displacement values of each measuring point is often complex and uneven [12], which is difficult to accurately express by fitting multiple terms of spatial coordinates only. (2) The spatial correlation between multiple measuring points is not constant, but varies from moment to moment [13], and it is difficult to reflect this variability using static spatial coordinates of measuring points. (3) Combining multiple terms of measuring point coordinates with environmental causal factors results in a model with too many input variables and too much model complexity [9].

In recent years, with the rapid development of deep learning theory, some scholars are trying to introduce deep learning models with better fitting ability into the field of dam deformation prediction [14,15], compared with machine learning models and shallow artificial neural networks, deep learning models can fully consider the lag effect of causal factors (i.e., the influence of causal factors at historical moments on the deformation of measuring points). A Convolutional Neural Network (CNN) is one of the most important networks in deep learning with high spatio-temporal feature extraction capability and computational efficiency, which has been widely used in remote sensing image recognition [16,17], traffic flow prediction [18,19], intelligent road tunnel investigations [20] and other fields, but less applied in the field of dam deformation prediction.

In summary, in order to improve the problems of the multi-point model with fused spatial coordinates and improve the prediction accuracy of the model, this paper proposes a 2D-1D-CNN model with fused spatial feature vectors by combining convolutional neural networks, takes a concrete dam as an example and compares the prediction results of this model with the current commonly used models to verify its effectiveness. Meanwhile, the prediction results of the SVM model are used as samples to explore the differences in prediction performance between the multi-point model with fused spatial coordinates and the multi-point model with fused spatial feature vectors.

## 2. 2D-1D-CNN Deformation Prediction Model for Concrete Dams

### 2.1. Multi-Point Model of Concrete Dams Incorporating Spatial Coordinates

Based on the existing theory of dam construction [1], this paper mainly considers the deformation of the dam body caused by water pressure, temperature, and aging. Among

them, the deformation caused by water pressure and temperature is reversible elastic deformation, and the deformation caused by aging factors is irreversible deformation caused by the deterioration of the dam material. Therefore, the concrete dam deformation can be divided into the water pressure component $\delta_H$, the temperature component $\delta_T$ and the aging component $\delta_\theta$, as shown in Equation (1):

$$\delta = \delta_H + \delta_T + \delta_\theta \tag{1}$$

For concrete dams in which the heat of hydration has been completely dissipated during the operation period, the harmonic factor can be chosen as the temperature factor. Therefore, Equation (2) can be used as the expression of the concrete dam deformation model [8].

$$\delta = a_0 + \sum_{i=1}^{4} a_i H^i + \sum_{i=1}^{2} \left( b_{1i} \sin \frac{2\pi i \Delta t}{365} \right) + \sum_{i=1}^{2} \left( b_{2i} \cos \frac{2\pi i \Delta t}{365} \right) + c_1 \theta + c_2 \ln \theta \tag{2}$$

where $H$ is the upstream water level height; $\Delta t$ is the cumulative number of days from the deformation monitoring date to the starting monitoring date; $\theta = \Delta t / 100$; $a_0$ is a constant term; $a_i$, $b_{1i}$, $b_{2i}$ and $c_1$, $c_2$ are all fitting coefficients.

Since the deformation of concrete dams exists in both time and space dimensions, the traditional single-point statistical model only models a single-point, which has difficulty in accurately reflecting the overall deformation state of the dam. The multi-point model of concrete dams is based on the single-point model, and the spatial coordinates of the measuring points are combined with the spatial correlation between the deformation sequences of measuring points, so it can better reflect the overall deformation of the dam. By incorporating the spatial coordinates of the measuring points $(x, y, z)$ into the single-point model of the concrete dam, the expression of the multi measuring point model of the concrete dam can be derived, as shown in Equations (3) and (4) [9].

$$\delta = f_1[(f(H), f(x, y, z)] + f_2[(f(T), f(x, y, z)] + f_3[(f(\theta), f(x, y, z)] \tag{3}$$

where $H$, $T$, $\theta$ are water pressure, temperature, and aging factor, respectively.

$$f(x, y, z) = \sum_{l,m,n=0}^{3} A_{lmn} x^l y^m z^n \tag{4}$$

Jointly with (2), the expression of the concrete dam multi-point model can be expressed as

$$\begin{aligned}
\delta = & \sum_{i=0}^{4} \sum_{l,m,n=0}^{3} A_{ilmn} H^i x^l y^m z^n + \\
& \sum_{i,j=0}^{1} \sum_{l,m,n=0}^{3} B_{ilmn} \sin \frac{2\pi i \Delta t}{365} \cdot \cos \frac{2\pi j \Delta t}{365} x^l y^m z^n + \\
& \sum_{i,j=0}^{1} \sum_{l,m,n=0}^{3} C_{ilmn} \theta_i \ln \theta_j x^l y^m z^n
\end{aligned} \tag{5}$$

where $A_{lmn}$, $B_{lmn}$ and $C_{lmn}$ are the fitting coefficients.

### 2.2. Convolutional Neural Network

As one of the most important deep learning frameworks, a convolutional neural network has efficient feature extraction ability and a lower number of parameters [21], and can be divided into 3D-CNN, 2D-CNN and 1D-CNN according to the application object. 2D-CNN is commonly used for the spatial feature extraction of panel data, and 1D-CNN is commonly used for the temporal feature extraction of sequences. A convolutional neural network mainly consists of a convolutional layer, pooling layer and fully connected layer.

Among them, the convolutional layer is used to extract data features, and the calculation process is shown in Equation (6).

$$X_{conv} = f(X * filter + b) \tag{6}$$

where $X$ is the input data, $*$ represents the convolution operation, $filter$ is the convolution kernel, $b$ is the bias, $f$ is the activation function and $X_{conv}$ is the output data of the convolution layer.

The pooling layer is used to extract key features from the data, which can be divided into maximum pooling and average pooling to reduce the computational effort of the model while reducing the risk of overfitting. The fully-connected layer is used to integrate the local feature vectors extracted by the convolutional and pooling layers to form features with practical meaning.

### 2.3. 2D-1D-CNN Concrete Dam Deformation Prediction Model Construction Process

The literature [22] pointed out that the spatial correlation between the measuring point deformation sequences is manifested by the correlation between the change patterns of the displacement values of each measuring point within the same period, and, on this basis, the spatial correlation of the measuring point deformation is quantified by the time series similarity. In this paper, based on work in the literature [22], a 2D-1D-CNN model is constructed by combining the idea of spatio-temporal data modeling and extracting the spatial features among measuring points through convolutional neural networks. The model consists of three parts, 2D-CNN, 1D-CNN and a tensor calculation layer. Among them, the 2D-CNN part extracts the spatial feature vector $\mu$, which can reflect the spatial features of each moment, from the historical deformation sequence of multiple measuring points, and replaces the spatial coordinates in the traditional multiple measuring point model with it. The tensor calculation layer refers to the expression of the traditional multi-point model and combines the spatial feature factor with the causal factors of the dam deformation to obtain the final input factors. Considering the advantages of 1D-CNN in sequence prediction with fewer parameters and higher prediction accuracy [23], 1D-CNN is selected in this paper to construct the nonlinear relationship between input factors and measuring point deformation to obtain the final prediction results. The specific structure of the 2D-1D-CNN model is shown in Figure 1.

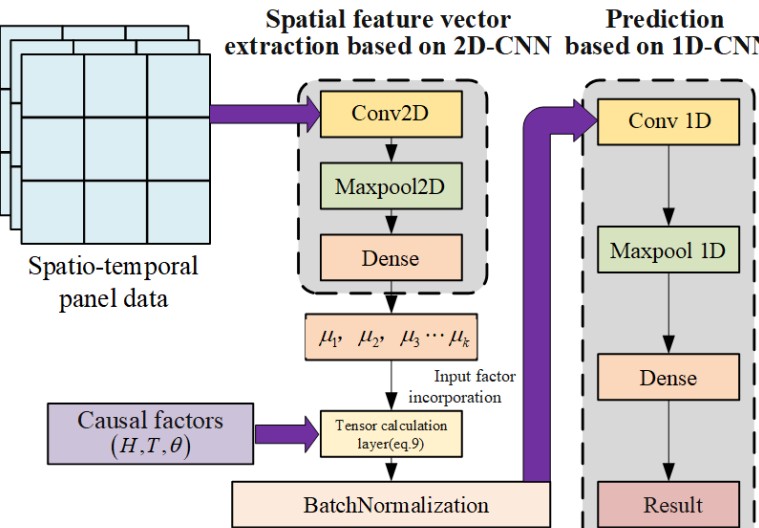

**Figure 1.** Schematic diagram of 2D-1D-CNN model.

The model expression is shown in Equation (7).

$$
\begin{aligned}
\delta = & f_1[f(H), f(\mu)] + f_2[f(T), f(\mu)] + f_3[f(\theta), f(\mu)] \\
= & \sum_{i=1}^{4} \sum_{p=1}^{k} A_{ilmn} a_i H^i \mu_p + \\
& \sum_{i,j=0}^{1} \sum_{p=1}^{k} B_{ilmn} \sin \frac{2\pi i \Delta t}{365} \cdot \cos \frac{2\pi j \Delta t}{365} \mu_p + \\
& \sum_{i,j=0}^{1} \sum_{p=1}^{k} C_{ilmn} \theta_i \ln \theta_j \mu_p
\end{aligned}
\tag{7}
$$

where $k$ is the dimension of the spatial feature vector, which belongs to the hyper parameters of the model.

The specific steps for constructing the model are as follows:

(1) Data set construction

As shown in Figure 2, each measuring point is constructed into a spatio-temporal panel data in the form of a grid according to its relative spatial position, with each small square representing the relative position of a measuring point, followed by filling the displacement values of each measuring point into the corresponding square, and filling the part of missing measuring points with 0. According to the above process, the deformation sequence of measuring points on the dam is constructed into a three-dimensional matrix of dimension $X_h \in R^{S1 \times S2 \times t}$ as the panel data input of the model, where $S1$, $S2$, represent the horizontal and vertical numbers of measuring point distribution, and $t$ denotes the length of the time window.

According to Equation (2), this paper takes $H, H^2, H^3, H^4, \sin \frac{2\pi \Delta t}{365}, \cos \frac{2\pi \Delta t}{365}, \sin \frac{4\pi \Delta t}{365}, \cos \frac{4\pi \Delta t}{365}, \theta, \ln \theta$ as the environmental casual factors of dam deformation, and constructs sliding windows in the time dimension for data addition, followed by the REPEAT function. The environment casual factor is replicated and filled n times ($n$ is the number of measuring points), and, finally, the casual factor input $X_f \in R^{t \times n \times 10}$ is obtained.

(2) Spatial feature vector extraction

The spatio-temporal panel data $X_h \in R^{S1 \times S2 \times t}$ in (1) is input to the 2D-CNN, and the spatial feature vector $\mu \in R^{t \times n \times k}$, which can reflect the spatial features at the current moment, is finally obtained through convolution calculation, and the calculation process is shown in Equation (8).

$$
\mu = W_1 \cdot MaxPool\{Relu[Conv2D(X_t)]\} + b_1
\tag{8}
$$

where $X_t$ is the panel data input of the model, $W_1$, $b_1$ are the weights and bias, MaxPool represents the maximum pooling operation and Relu is the activation function.

(3) Input factors incorporation

The tensor calculation layer is constructed to fuse $\mu = (\mu_1, \mu_2, \ldots \mu_k)$ with the environmental casual factors input according to Equation (9), and the fused factors are normalized by the Batch Normalization layer to obtain the final input factors $X_{f'} \in R^{t \times n \times (k \times 10)}$ of the fused spatial feature vector.

$$
X_{f'} = BatchNorm\left[concat(\mu_1 \cdot X_f, \mu_2 \cdot X_f, \cdots, \mu_k \cdot X_f)\right]
\tag{9}
$$

where *BatchNorm* represents the batch normalization process and *concat* represents the tensor stitching operation.

(4) Concrete dam deformation prediction

$X_{f'}$ is input to 1D-CNN for prediction, and the final prediction results are obtained.

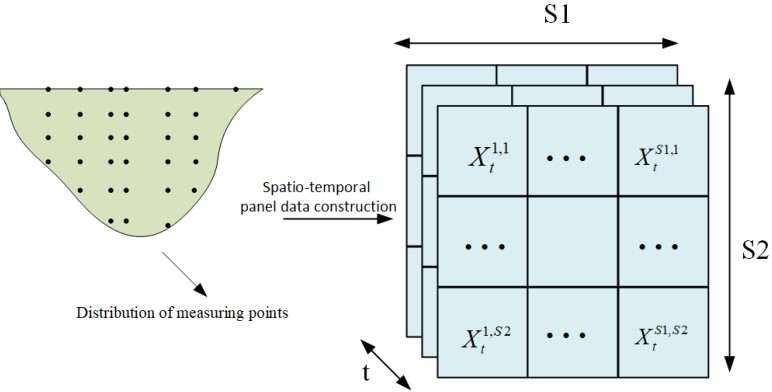

**Figure 2.** Schematic diagram of the spatio-temporal panel data construction process.

## 3. Case Study

Based on the above method, a 2D-1D-CNN model is constructed for concrete dam deformation prediction based on the actual monitoring data of a hydropower station in the south of China, which has a maximum dam height of 67.5 m and a total installed capacity of $4 \times 18{,}000$ kW. The project profile is shown in Figure 3. In this paper, the deformation data from 1 January 2009 to 22 October 2010 (shown in Figure 4) were selected from measuring points EX3-EX7 in the dam crest tension line, and the sampling interval was once every 5 days. Root Mean Square Error (RMSE), Mean Absolute Error (MAE) and Mean Square Error (MSE) were used as statistical indexes.

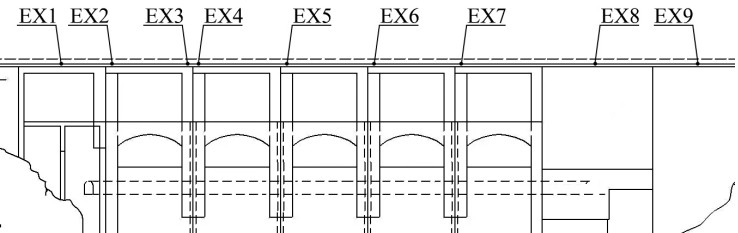

**Figure 3.** Dam project overview map.

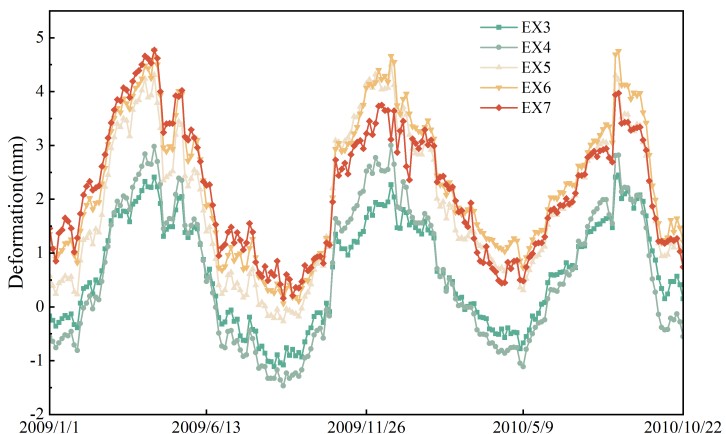

**Figure 4.** Deformation sequence of measuring points.

### 3.1. Model Activation Functions Selection

The ReLU activation function is chosen for the proposed model in this paper. In order to verify the effect of different activation functions on the model, we constructed a 2D-1D-CNN model based on the sigmiod activation function, and analyzed the effect of different activation functions on the prediction performance of the model through the

change of the loss curves of these two models during the training process, as shown in Figure 5.

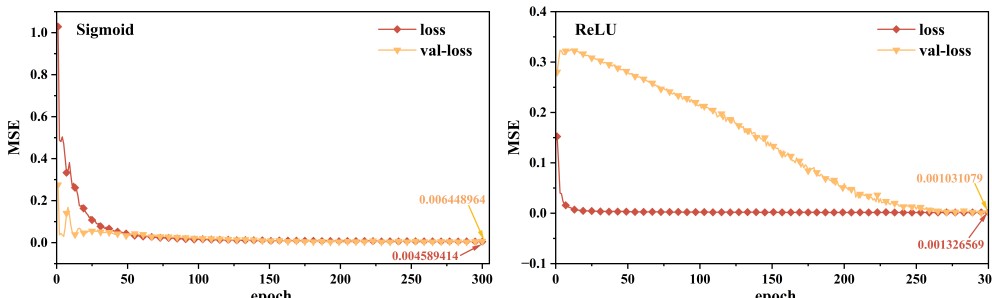

**Figure 5.** Comparison of the model's loss curves with different activation functions.

Figure 2 shows that the 2D-1D-CNN model is not sensitive to the choice of activation function, and the loss values can converge better in both cases. Overall, when choosing the sigmoid activation function, the loss value of the model test set converges faster and, when choosing the ReLU activation function, the loss value of the model test set can drop lower, which also means higher prediction accuracy. Therefore, the ReLU activation function is preferred in this paper.

### 3.2. Model Hyperparameters Selection

In order to investigate the influence of the spatial feature factor dimension k on the prediction accuracy of the model, the prediction was carried out in the cases of spatial feature factor 1–6 dimensions, and the optimal k value was determined by comparing the average value of the RMSE index of each measuring point, and the results are shown in Table 1.

**Table 1.** RMSE index predicted by the model under different k values.

| k | 1 | 2 | 3 | 4 | 5 | 6 |
|---|---|---|---|---|---|---|
| RMSE | 0.2291 | 0.2029 | 0.1688 | 0.1468 | 0.1720 | 0.2085 |

Table 1 shows that, when k is taken as 4, the model has the best prediction effect, so k = 4 is taken in this paper. At the same time, after several experimental analyses, the time window was set to 6, the convolution kernel size in 2D-CNN was set to [1, 3], the sliding step size was set to 1, and the number of convolution kernels was set to 96. In order to reduce the calculation amount of the model, the size of the first layer of convolution kernel size in 1D-CNN is set to 3, the dilatation rate is set to 2, the size of the second layer of the convolution kernels is set to 2, and the number of convolution kernels is uniformly set to 64. The model is built based on Tensorflow 2.10, the loss function is selected as MSE, the optimizer is selected Adam, the batch size is set to 32, the number of epochs is set to 300 and the learning rate is 0.001.

### 3.3. Prediction Results and Comparative Analysis

To comprehensively evaluate the prediction performance of the proposed model, the prediction results are compared with the SVM multi-point model with fused spatial coordinates, single-point BP neural network, a linear regression model based on $HT_AT$ and a single-point 1D-CNN model; the prediction results are shown in Figure 6. In addition, since this paper only models and analyzes the horizontal displacement of the measuring points on the lead tension line, only the coordinate changes in the x-direction are considered in the fused spatial coordinates in this paper.

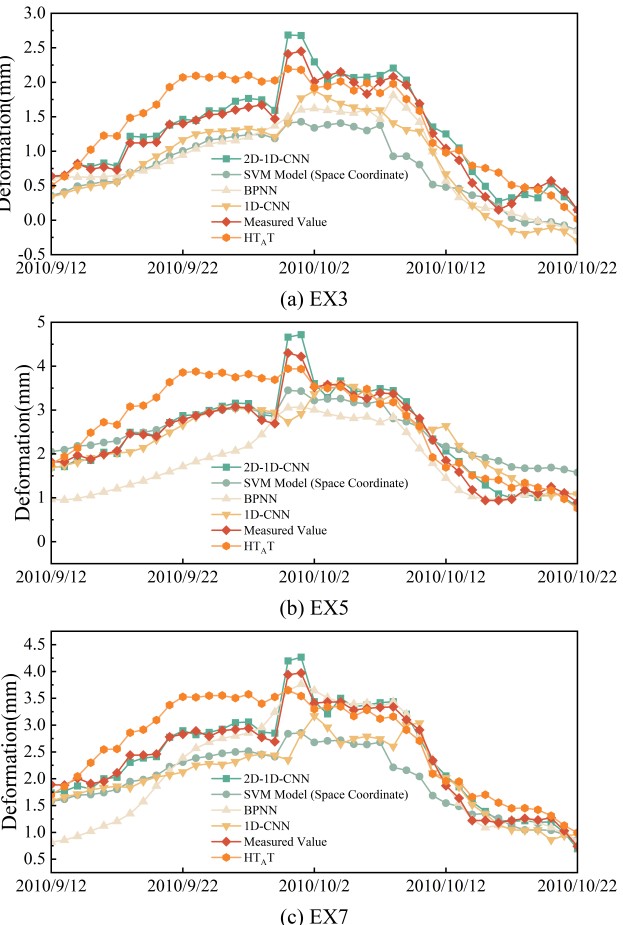

**Figure 6.** Comparison of model prediction results.

Figure 6 shows that the prediction results of the proposed model at the three measuring points can better fit the measured values, and the prediction results can better fit the real displacement trend at the measuring points than the prediction results of other comparative models, which also shows that the proposed model can accurately reflect the overall deformation behavior of the dam. In addition, the HT$_A$T-based linear regression model has a higher prediction accuracy in the part where the deformation fluctuates locally, but deviates more for the part where the deformation is more gentle. Overall, the advantages of introducing air temperature are not obvious for this study case. In order to show the model prediction accuracy more intuitively, the prediction accuracy indexes of each model at the three measuring points were calculated, as shown in Table 2.

Table 2 shows that the average RMSE index of the proposed model in this paper achieves 72.6%, 74.58% and 69.89% relative to the single-point BPNN model, multi-point SVM model and 1D-CNN, respectively, indicating that the proposed model is reasonable and effective. Compared with the comparison models, the prediction accuracy of the proposed model is higher, which is mainly because the 2D-1D-CNN model models multiple measuring points at the same time, considering the interconnection of displacement values among multiple measuring points on the one hand, and considering the deformation sequences of multiple measuring points while, at the same time, also expanding the training samples of the model to a certain extent. In addition, the deep learning model can fully consider the influence of the environmental factors on the deformation of the measuring points at the historical time (i.e., the lag effect of the causal factors) and the change law of the spatial association of the deformation of the measuring points with time (i.e., the change characteristic of the spatial association of multiple measuring points), so its prediction accuracy is higher than that of the machine-learning-based model and the shallow artificial neural network in most cases.

**Table 2.** Model prediction statistical indexes comparison table.

| Monitoring Points | Statistical Indexes | 2D-1D-CNN | SVM | BPNN | 1D-CNN | HT$_A$T |
|---|---|---|---|---|---|---|
| | RMSE/mm | 0.121 | 0.5472 | 0.4162 | 0.4222 | 0.3426 |
| EX3 | MAE/mm | 0.0971 | 0.4782 | 0.3649 | 0.3826 | 0.2840 |
| | MAPE | 11.75% | 46.82% | 39.14% | 51.27% | 38.30% |
| | $R^2$ | 0.9658 | 0.8678 | 0.5957 | 0.5840 | 0.7261 |
| ine | RMSE/mm | 0.1635 | 0.4263 | 0.7347 | 0.4413 | 0.5345 |
| EX5 | MAE/mm | 0.1241 | 0.3245 | 0.6284 | 0.2713 | 0.4236 |
| | MAPE | 6.47% | 21.04% | 24.90% | 13.82% | 19.43% |
| | $R^2$ | 0.9694 | 0.8486 | 0.3833 | 0.7775 | 0.6736 |
| ine | RMSE/mm | 0.1296 | 0.5349 | 0.4781 | 0.5115 | 0.4134 |
| EX7 | MAE/mm | 0.1045 | 0.4501 | 0.3176 | 0.412 | 0.3437 |
| | MAPE | 4.95% | 17.27% | 14.53% | 16.54% | 15.94% |
| | $R^2$ | 0.9772 | 0.8817 | 0.6892 | 0.6442 | 0.7676 |

*3.4. Validation of the Validity of a Multi-Point Model Incorporating Spatial Feature Vectors*

To more intuitively verify the validity of the multi-point model with fused feature vectors, the final input factor $X_{f'}$ constructed in the 2D-1D-CNN model in Section 3.2 is extracted from the network, reconstructed as the input factor $X_{f''} \in R^{n \times (k \times 10)}$ with fused spatial features according to the format of the input factor of the SVM model, input to the SVM model prediction and compared with the conventional multi-point SVM model with the introduction of spatial coordinates in Section 3.2 to analyze the variation of prediction accuracy of the SVM model with different input factors; the results are shown in Figure 7 and Table 3.

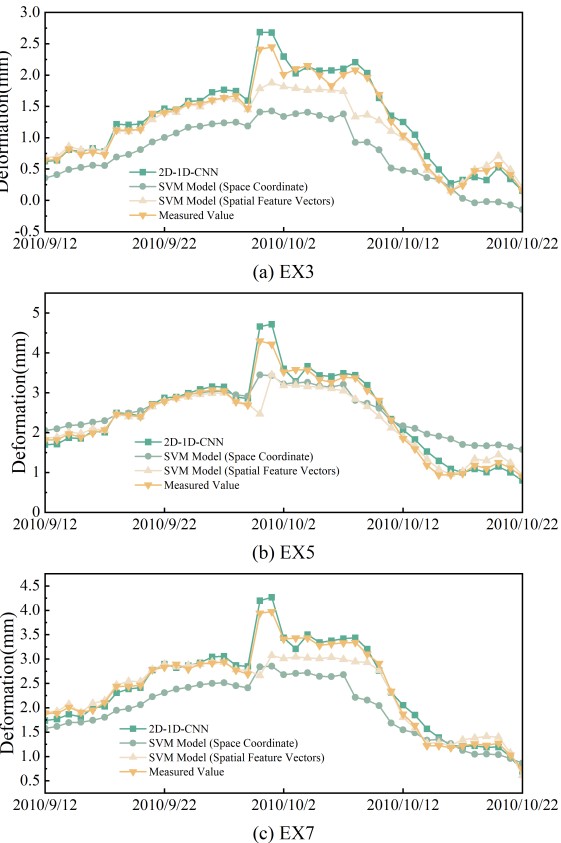

**Figure 7.** Comparison of SVM prediction results with different input factors.

**Table 3.** Comparison of SVM prediction statistical indexes with different input factors.

| Monitoring Points | Statistical Indexes | 2D-1D-CNN | SVM (Space Coordinate) | SVM (Spatial Feature Vector) |
|---|---|---|---|---|
| | RMSE/mm | 0.121 | 0.5472 | 0.238 |
| EX3 | MAE/mm | 0.0971 | 0.4782 | 0.1406 |
| | MAPE | 11.75% | 46.82% | 10.52% |
| | $R^2$ | 0.9658 | 0.3010 | 0.8678 |
| ine | RMSE/mm | 0.1635 | 0.4263 | 0.364 |
| EX5 | MAE/mm | 0.1241 | 0.3245 | 0.1946 |
| | MAPE | 6.47% | 21.04% | 7.45% |
| | $R^2$ | 0.9694 | 0.7923 | 0.8486 |
| ine | RMSE/mm | 0.1296 | 0.5349 | 0.2949 |
| EX7 | MAE/mm | 0.1045 | 0.4501 | 0.1671 |
| | MAPE | 4.95% | 17.27% | 6.42% |
| | $R^2$ | 0.9772 | 0.6110 | 0.8817 |

Figure 7 and Table 3 show that the prediction results of the SVM model with spatial feature vectors can be more in line with the trend of measured values, and the prediction accuracy is more significantly improved at the moment when the deformation fluctuation is large. The RMSE index on the three measuring points decreased by 56.51%, 14.62% and 44.87%, respectively. This shows that the spatial feature vectors used in this paper can more accurately reflect the spatial correlation between the deformation sequences of each measuring point and have validity. At the same time, because the traditional multi-point model needs to combine the causal factor with the zeroth to third degree of the spatial coordinates, every time the one-dimensional coordinate increases, the number of input factors increases exponentially by four, which easily produces the problem of excessive input factors. The spatial feature factor constructed in this paper reflects the spatial characteristics in the form of vectors, and the complexity of the space increases linearly, so the overall computational efficiency is stronger than that of the traditional multi-point model; with the increase of the coordinate dimension, the advantages of the spatial feature factor proposed in this paper in computational efficiency will be more prominent.

## 4. Conclusions

In this paper, combined with a convolutional neural network, a dam deformation prediction model (2D-1D-CNN) based on spatial feature vectors is proposed. After an experimental analysis of the measured data of a concrete dam, the following conclusions are obtained:

(1) The multi-point model based on the fusion of spatial feature vectors can effectively extract the spatial connection between the measuring points and, compared with the other models, the model proposed in this paper shows a great improvement in the prediction accuracy.In addition, the experimental results indicated that the prediction accuracy of the deep-learning-based prediction model is higher than that of the machine learning model and the statistical model in most cases.

(2) The case study shows that the proposed 2D-1D-CNN model is able to accurately predict the deformation of concrete dams so that it is effective and feasible.

(3) In this paper, 2D-CNN and 1D-CNN are fused into the same network model, which, on the one hand, improves the fitting ability of the network for dam deformation sequences, and, on the other hand, it also increases the hyperparameters that need to be adjusted to a certain extent. Since this paper only adopts the artificial coarse tuning method, the optimization algorithm can be combined in future studies to further improve the prediction performance of the model.

**Author Contributions:** Conceptualization, Z.C.; methodology, Z.C.; writing—original draft preparation, Z.C.; writing—review and editing, X.L. All authors have read and agreed to the published version of the manuscript.

**Funding:** This research was funded by the National Natural Science Foundation of China (grant numbers 42171437).

**Institutional Review Board Statement:** Not applicable.

**Informed Consent Statement:** Not applicable.

**Data Availability Statement:** Not applicable.

**Conflicts of Interest:** The authors declare no conflict of interest.

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
