# Peer review of "Multi-Point Deformation Prediction Model for Concrete Dams Based on Spatial Feature Vector"

_applsci, doi:10.3390/app132011212_

Round 1

Reviewer 1 Report

The manuscript "Multi-point deformation prediction model for concrete dams based on spatial feature vector" is new, original and interesting. The authors combined with convolutional neural network, a dam deformation prediction model (2D-1D-CNN) based on spatial feature vectors is proposed. The case study shows that the proposed 2D-1D-CNN model is able to accurately predict the deformation of concrete dams so that it is effective and feasible. My recommendation is accept as is.

Author Response

Thank you very much for taking the time to review this manuscript and for acknowledging it.

Reviewer 2 Report

Manuscript ID applsci-2624370 entitled "Multi-point deformation prediction model for concrete dams based on spatial feature vector" has been reviewed, the study takes into consideration the importance of dam deformation monitoring in ensuring structural safety and proposes a 2D-1D-CNN dam deformation monitoring and prediction model for concrete dam, the manuscript is well-structured, clearly articulated, and the proposed model demonstrates a certain level of innovation, as well as better practical applicability. However, the following minor revision recommendations were provided as follows to enhance the clarity, accuracy, and readability of the manuscript, in alignment with the requirements of Applied Science Journal.

(1)   Some expressions in the manuscript are not entirely appropriate and have been highlighted in yellow; they require modification. The English writing in the manuscript requires some levels of polishing and revision.

(2)   The numerical labeling in Figure 3 needs to be clearer and more standardized.

(3)   The fourth paragraph of the introduction explains three shortcomings of the spatial coordinate-based multi-measurement point prediction model, of which the first and third points do not cite relevant literature.

(4)   The article constructs a dam deformation prediction model based on the deep learning model, which should be supplemented in the conclusion to illustrate the advantages of the deep learning model compared with machine learning and statistical models.

(5)   The part of section 2.2 that introduces the principle of convolutional neural network needs to cite relevant literature.

Some expressions in the manuscript are not entirely appropriate and have been highlighted in yellow; they require modification. The English writing in the manuscript requires some levels of polishing and revision.

Author Response

Thank you very much for taking the time to review this manuscript. Please find the detailed responses below and the corrections highlighted in the re-submitted files.

(1)   Some expressions in the manuscript are not entirely appropriate and have been highlighted in yellow; they require modification. The English writing in the manuscript requires some levels of polishing and revision.

Response :Revised, we have changed some English words and rewritten some sentences to make them more accurate,please see line 6,9,51,256-258.

(2)   The numerical labeling in Figure 3 needs to be clearer and more standardized.

Response :Revised, please see Figure 3.

(3)   The fourth paragraph of the introduction explains three shortcomings of the spatial coordinate-based multi-measurement point prediction model, of which the first and third points do not cite relevant literature.

Response :Added, please see line 44 and line 49.

(4)   The article constructs a dam deformation prediction model based on the deep learning model, which should be supplemented in the conclusion to illustrate the advantages of the deep learning model compared with machine learning and statistical models.

Response :Added, please see line 44 and line 49.

(5)   The part of section 2.2 that introduces the principle of convolutional neural network needs to cite relevant literature.

Response :Added, please see line 99.

Reviewer 3 Report

The paper is indeed engaging and well-written. However, there are several key aspects that should be addressed before considering it for publication:

  1. Missing Reference: There is a missing reference in the sentence: "Therefore, equation (2) can be used as the expression of the concrete dam deformation model [? ]." It's crucial to provide the appropriate citation to support this statement.

  2. Choice of Activation Function: The authors have chosen the ReLU activation function. It would be valuable to understand the sensitivity of the model to this choice. Consider including an analysis that explores the impact of alternative activation functions such as Sigmoid. This would provide insights into the robustness of the model.

  3. Inclusion of R^2 and Predicted vs. Measured Values: Adding the coefficient of determination (R^2) as a performance indicator is advisable. Furthermore, it would be beneficial to include a graphical representation of predicted vs. measured values. This visual comparison can help assess the model's accuracy and predictive power.

  4. Practical Significance of Input Variables: The paper currently focuses on time and space as input explanatory variables. To enhance the comprehensiveness and practical relevance of the research, consider incorporating temperature as an additional input variable. Additionally, provide the results of linear regression on temperature to illustrate how well or poorly temperature alone serves as an explanatory variable.

  5. Expand the Reference List: To strengthen the paper's scholarly foundation, consider including the following references in the bibliography:

    • Rosso, M. M., Marasco, G., Aiello, S., Aloisio, A., Chiaia, B., & Marano, G. C. (2023). "Convolutional Networks and Transformers for Intelligent Road Tunnel Investigations." Computers & Structures, 275, 106918.

    • Hua, G., Wang, S., Xiao, M., & Hu, S. (2023). "Research on the Uplift Pressure Prediction of Concrete Dams Based on the CNN-GRU Model." Water, 15(2), 319.

Author Response

Thank you very much for taking the time to review this manuscript. Please find the detailed responses below and the corrections highlighted in the re-submitted files.

  1. Missing Reference:There is a missing reference in the sentence: "Therefore, equation (2) can be used as the expression of the concrete dam deformation model [? ]." It's crucial to provide the appropriate citation to support this statement.

Response :Added, please see line 80.

2.Choice of Activation Function: The authors have chosen the ReLU activation function. It would be valuable to understand the sensitivity of the model to this choice. Consider including an analysis that explores the impact of alternative activation functions such as Sigmoid. This would provide insights into the robustness of the model.

Response :Added, please see Section 3.1

3.Inclusion of R^2 and Predicted vs. Measured Values: Adding the coefficient of determination (R^2) as a performance indicator is advisable. Furthermore, it would be beneficial to include a graphical representation of predicted vs. measured values. This visual comparison can help assess the model's accuracy and predictive power.

Response :Added, please see Tabel 2-Tabel 3 and Fig.6-Fig. 7.

4.Practical Significance of Input Variables: The paper currently focuses on time and space as input explanatory variables. To enhance the comprehensiveness and practical relevance of the research, consider incorporating temperature as an additional input variable.

Response :Due to the large structure of dams, a large number of thermometers need to be built into the dam in order to more accurately measure the temperature in the area where each monitoring point is located. Unfortunately, due to limited conditions, many dams are not constructed with a sufficient number of buried thermometers (including the case studied in this paper). Therefore existing methods generally approximate the effect of temperature on dam deformation by averaging the temperature or constructing a harmonic function(  ï¼‰.

Literature [1] points out that for dams with stable service ambient temperature and complete hydrothermalization emanation, the temperature of the dam concrete varies with seasonal evolution, so that the harmonic function can be used to approximately simulate the cyclic variation of temperature. Therefore, in this paper, the effect of temperature on dam deformation is modeled by using the harmonic function as a temperature factor, please see line 147.

Additionally, provide the results of linear regression on temperature to illustrate how well or poorly temperature alone serves as an explanatory variable.

Response :Literature [1] points out that the main factors of dam deformation are water pressure, temperature and time, so we believe that the deformation of dams cannot be fully explained by temperature changes alone.

In order to explore how good or bad the temperature change is as an explanatory variable, this paper constructs a linear regression model with input factors of water pressure, average air temperature, and ageing factor (HTAT) as a comparative model based on the literature [2] by replacing the internal temperature of the dam through air temperature, instead of a linear regression model with input factors of temperature, please see line 200 and line208-212.

5.Expand the Reference List: To strengthen the paper's scholarly foundation, consider including the following references in the bibliography:

Rosso, M. M., Marasco, G., Aiello, S., Aloisio, A., Chiaia, B., & Marano, G. C. (2023). "Convolutional Networks and Transformers for Intelligent Road Tunnel Investigations." Computers & Structures, 275, 106918.

Hua, G., Wang, S., Xiao, M., & Hu, S. (2023). "Research on the Uplift Pressure Prediction of Concrete Dams Based on the CNN-GRU Model." Water, 15(2), 319.

Response :References added, please see line 58 and line 99.

Literature [1]:Wei, B.; Liu, B.; Yuan, D.; Mao, Y.; Yao, S. Spatiotemporal hybrid model for concrete arch dam deformation monitoring considering chaotic effect of residual series. Eng. Struct. 2021, 228, 1114

Literature [2]:Kang, F.; Liu, X.; Li, J. Temperature effect modeling in structural health monitoring of concrete dams using kernel extreme learning machines. Struct. Health Monit. 2020, 19, 987–10

Round 2

Reviewer 3 Report

Accept